# Proteomic Analysis of Frozen–Thawed Spermatozoa with Different Levels of Freezability in Dairy Goats

**DOI:** 10.3390/ijms242115550

**Published:** 2023-10-25

**Authors:** Pingyu Sun, Guoyu Zhang, Ming Xian, Guangzhi Zhang, Fei Wen, Zhangtao Hu, Jianhong Hu

**Affiliations:** Key Laboratory of Animal Genetics, Breeding and Reproduction of Shaanxi Province, College of Animal Science and Technology, Northwest A&F University, Xianyang 712100, China; 18333002217@163.com (P.S.); zzr425983391@126.com (G.Z.); xianmingtz@126.com (M.X.); z13514592253@163.com (G.Z.); 13087561604@163.com (F.W.); huzhangtao0819@163.com (Z.H.)

**Keywords:** 4D-DIA, proteomics, dairy goats, sperm freezability

## Abstract

The results of artificial insemination (AI) are adversely affected by changes in sperm motility and function throughout the cryopreservation procedure. The proteome alterations of frozen–thawed spermatozoa with various levels of freezability in dairy goats, however, remain largely unknown. To discover differentially expressed proteins (DEPs) and their roles in dairy goat sperm with high or low freezability (HF or LF), we conducted 4D-DIA quantitative proteomics analysis, the results of which are presented in this work. Additionally, we explored the underlying processes that may lead to the variations in sperm freezing resistance. A total of 263 DEPs (Fold Change > 2.0, *p*-value < 0.05) were identified between the HF group and LF group in frozen–thawed dairy goat spermatozoa. In our Gene Ontology (GO) enrichment analysis, the DEPs were mostly associated with the regulation of biological processes, metabolic processes, and responses to stress and cellular component biogenesis. Our Kyoto Encyclopedia of Genes and Genomes (KEGG) analysis also revealed that the DEPs were predominantly engaged in oxidative phosphorylation, N-Glycan biosythesis, and cysteine and methionien metabolism. A protein–protein interaction (PPI) network analysis revealed 14 potential proteins (NUDFB8, SDHC, PDIA4, HSPB1, etc.) that might influence the freezability of dairy goat sperm. These findings shed light on the processes underlying alterations in the proteome and sperm freezability, aiding further research on sperm cryopreservation.

## 1. Introduction

The construction of sperm biobanks, the preservation of genetic resources, and goat breeding programs all involve the preservation of sperm. The peculiar plasma membrane structure of goat sperm and many circumstances, such as the freezing process and cryopreservation extenders, might, however, affect the quality of frozen semen. As a result, problems such as reduced motility, a lower fertilization rate, and a very low utilization rate in output may arise [1,2]. Prior research has shown that the quality of frozen semen differs not just between various breeds but also between individuals within particular breeds [3,4,5]. In recent years, a greater emphasis has been placed on protein levels, with proteins like heat shock protein 90 kDa alpha A1 (HSP90AA1) [6] and enolase 1 (ENO1) [7] being identified as potential biomarkers for sperm freezing resistance. These protein variations can influence spermatozoa cryopreservation in a variety of ways, making them useful indicators for predicting sperm freezability.

How protein breakdown during cryopreservation affects the architecture and functions of sperm has been thoroughly investigated [4]. A potential method for examining the mechanism of sperm cryoinjury is proteome analysis. Numerous researchers have investigated the relationship between changes in protein profiles and sperm quality [6,7,8]. According to Casas I, heat shock protein 90 kDa alpha A1 (HSP90AA1) protein was found to be significantly more abundant in frozen sperm than in fresh sperm. Sperm freezing resistance was discovered to be correlated with the expression levels of HSP90AA1 [6]. Based on research regarding the freezing resistance of human semen, it was observed that spermatozoa with high freezing resistance exhibited much greater amounts of the enzymes enolase 1 (ENO1) and glucose 6-phosphate isomerase (GPI) compared to spermatozoa with poor freezing resistance [7]. Similarly, in goats, sperm freezing resistance was favorably linked with the 26S proteasome complex, HSP90AA1, and transitional endoplasmic reticulum ATPase [8].

The effect of the freezing and thawing process on the proteome of goat sperm is not well understood, especially when compared to other livestock species like cattle, pig, and sheep. This is true despite the abundance of papers on sperm proteomics. Thus, it is essential to look at how cryopreservation affects the proteome of goat sperm. Previous research has shown that goat sperm’s susceptibility to the stress of freezing and thawing varies [9]. For the current study, we conducted 4D-DIA quantitative proteomics analysis of different freezing resistant spermatozoa from dairy goats. The goal of this integrated proteome analysis was to thoroughly examine the proteins associated with the capacity of dairy goat spermatozoa to be frozen. Additionally, we sought to advance the cryopreservation of dairy goat spermatozoa by offering fresh perspectives and laying a theoretical framework.

## 2. Results

### 2.1. Evaluation of Sperm Quality

A decline in total motility was observed in all of the goats after subjecting the spermatozoa to the freezing–thawing process. However, the extent of this decline varied among the 12 goats. Based on their total motility, these goats were categorized into two groups: the HF group, consisting of goats with more than 75% total motility (*n* = 4), and the LF group, consisting of goats with less than 45% total motility (*n* = 4) (Figure 1). CASA analysis was performed to assess different kinetic parameters, such as TM, PM, VAP, VCL, VSL, and LIN. These parameters exhibited significant differences between the HF and LF groups (*p* < 0.001) (Table 1). In order to determine the level of oxidative damage present in the membrane, it is important to assess the plasma membrane integrity. Additionally, the presence of an intact acrosome is crucial for the sperm acrosome reaction. The LF group exhibited a significant decrease in both sperm plasma membrane integrity and acrosomal integrity compared to the HF group, as shown in Figure 2 (*p* < 0.05).

### 2.2. Evaluation of Sperm Antioxidant Activity

Oxidative stress injury is the main cause of sperm quality decline during cryopreservation. Therefore, it is crucial to understand the antioxidant capacity of different freeze-resistant goat sperm (T-AOC, CAT, SOD activity, MDA, and ROS content). As shown in Figure 3, the T-AOC, CAT, and SOD activities of the LF group decreased significantly compared to the HF group; therefore, the levels of ROS and MDA were increased in the LF group (*p* < 0.05).

### 2.3. Quality Evaluation of Protein Extraction

In order to investigate the effect of different proteins on goat sperm during the freezing–thawing process, the quality of HF and LF sperm protein extraction was evaluated. The distribution of protein bands was normal, and a certain degree of high abundance of protein was present. The number of results regarding peptide sequence matching, analyzed via mass spectrometry, was 16,331, the protein number analyzed based on specific peptide segments was 2633, and the protein number quantified by specific peptide segments was 2607. The distributions of peptide length and number were determined (Figure 4).

### 2.4. Protein Identification Results

To evaluate whether the quantitative results of differential proteins in HF and LF sperm met the statistical concordance requirement, three statistical methods were employed. Pearson’s correlation coefficient analysis revealed that the quantitative results were consistent with statistical data, indicating good reproducibility between the two groups. The Principal Component Analysis (PCA) results demonstrated tight clustering and high reproducibility between the two groups. Additionally, the Relative Standard Deviation (RSD) results indicated small total RSD values between the two groups, further indicating good quantitative reproducibility (Figure 5).

### 2.5. Screening of DEPs

Using the fold change of protein expression differences between the HF and LF groups and the *p*-values obtained from the results of our *t*-test, a volcanic map was plotted to show the significant differences in sample data between the two groups (FC ≥ 2.0 or ≤0.5 and *p* < 0.05). A total of 263 DEPs (169 down-regulated proteins and 94 up-regulated proteins) were obtained from the frozen thawed samples (Appendix A). These DEPs were further subjected to function analysis and validation experiments (Figure 6).

### 2.6. Bioinformatics Analysis of DEPs

To categorize the DEPs, a GO analysis was performed. The DEPs were classified into three categories: biological process (BP), cellular component (CC), and molecular function (MF). Within the BP category, up-regulated DEPs were enriched in the following processes: the regulation of biological processes, the metabolic process of organic substances, the primary metabolic process, and responses to chemicals and stress. On the other hand, the down-regulated DEPs were enriched in the nitrogen compound metabolic process, the cellular metabolic process, cellular component organization or biogenesis, and the cellular response to stimulus. Moving on to the CC category, the up-regulated DEPs were found to be enriched in intracellular anatomical structures, cytoplasm, organelles, and membranes. Conversely, the down-regulated DEPs were enriched in membrane-enclosed lumen, extracellular regions, cytoplasm, organelles, and membrane. Lastly, for the MF terms, the up-regulated DEPs showed enrichment in protein binding, ion binding, hydrolase activity, and organic cyclic compound binding (Appendix A, Figure 7a). On the other hand, the down-regulated DEPs were enriched in heterocyclic compound binding, protein binding, organic cyclic compound binding, and heterocyclic compound binding (Appendix A, Figure 7c).

In order to identify the pathways that are enriched with DEPs, we performed KEGG enrichment analysis. This analysis successfully mapped a total of 263 DEPs to 56 term IDs. We observed that the up-regulated DEPs showed significantly high enrichment in various pathways, including phagosomes, gap junction, thyroid hormone synthesis, ABC transporters, complement and coagulation cascades, and the intestinal immune network for IgA production, among others (Appendix A, Figure 7b). On the other hand, down-regulated DEPs were found to be highly enriched in pathways related to nucleocytoplasmic transport, N-Glycan biosynthesis, ferroptosis, and spliceosomes and the mRNA surveillance pathway (Appendix A, Figure 7d).

### 2.7. Subcellular Localization of DEPs

The subcellular localization of up-regulated proteins after sperm thawing in the goats (HF vs. LF) was predominantly found in the extracellular regions (29.79%), cytoplasm (23.4%), nucleus (13.83%), plasma membrane (13.83%), and mitochondria (7.45%). Conversely, the proteins that were down-regulated after sperm thawing were largely associated with the cytoplasm (36.69%), nucleus (31.95%), extracellular regions (10.65%), and plasma membrane (5.92%) (Figure 8).

### 2.8. Protein–Protein Interaction (PPI) Analysis

In order to identify more key proteins related to critical pathways, a PPI network was built using STRING (confidence score > 0.7) (Figure 9). In the network we created, the highly connected DEPs were considered central proteins that may also play an important role in network regulation.

## 3. Discussion

The permeability of the sperm plasma membrane is impacted by sperm cryopreservation, and the acrosome integrity is compromised. In this study, it was discovered that HF sperm had greater levels of the following substances than LF sperm: TM, PM, VAP, VCL, VSL, and LIN. Additionally, HF sperm also had higher levels of antioxidant activity. These findings were used for further proteomics research. Our thorough examination of the data pertaining to proteomics allowed us to discover a total of 14 candidate proteins (NDUFB8, SDHC, COX6C, PDIA6, PDIA4, PPA1, FTH1, HSPB1, DNAJB1, ATP1A1, STIP1, SRSF1, ABCG1, ADAM7) that may have an effect on the capacity of dairy goat sperm to be frozen. These proteins could be involved in several pathways, such as oxidative phosphorylation, ferroptosis, oxidoreductase activity, and sperm capacitation, etc. This study is the first to report on the changes in the proteome of dairy goat spermatozoa caused by cryopreservation. It provides a new molecular perspective for investigating the mechanisms behind sperm damage resulting from the freezing and thawing process.

Energy metabolism is essential for sperm function and is specifically essential in maintaining sperm motility by providing the necessary ATP [10,11]. ATP production in mature spermatozoa primarily occurs through oxidative phosphorylation (OXPHOS), a process where substrate oxidation drives an electron transfer chain, resulting in the synthesis of ATP through an electrochemical transmembrane gradient. In eukaryotic cells, the OXPHOS pathway can generate significantly more energy (up to 90%). The relationship between sperm motility and energy metabolism is closely intertwined [12,13]. Four protein complexes that make up the respiratory chain, which are positioned on the inner membrane of the mitochondria, primarily support this process. These four complexes are complex I (NADH dehydrogenase), complex II (succinate coenzyme Q reductase), complex III (cytochrome C reductase), and complex IV (cytochrome C oxidase). According to our research, the DEPs’ enriched pathways mostly involve energy metabolism. Notably, three major genes—NDUFB8, SDHC, and COX6C—were found to be up-regulated simultaneously and played significant roles in OXPHOS. This finding differs from that of a previous study [14]. NDUFB8 is a part of the mitochondrial respiratory chain complex I and contributes to its construction. The subunit of succinate dehydrogenase known as SDHC is also known as mitochondrial complex II. This enzyme complex is essential to the mitochondria’s aerobic respiratory chain. One of the essential components of COX6C is the mitochondrial respiratory chain’s terminal enzyme. Previous research has shown that COX6C is crucial for controlling OXPHOS and generating energy [15]. In the mitochondrial aerobic respiratory chain, these DEPs are important enzyme complexes that transfer electrons and provide energy to sperm. Previous studies have shown that blocking the mitochondrial respiratory chain in sperm results in a marked reduction in the parameters governing sperm motility in bovine [16]. Additionally, sperm ATP levels and motility are diminished when human semen is treated with mitochondrial respiratory chain inhibitors [17]. These results imply that sperm motility depends on ATP synthesis via OXPHOS. In this study, it was discovered that the HF group had up-regulated levels of NDUFB8, SDHC, and COX6C. The enrichment analysis results also showed that the main players in the OXPHOS process were NDUFB8, SDHC, and COX6C. Therefore, we propose that during sperm freezing–thawing, NDUFB8, SDHC, and COX6C produce ATP via oxidative phosphorylation, supplying energy for sperm motility and perhaps affecting sperm freezing tolerance. PPA1 (inorganic pyrophosphatase 1) was shown to be considerably down-regulated during the freeze–thawing of goat spermatozoa. This enzyme was significantly enriched in pathways associated with oxidative phosphorylation and hydrolase activity. Inorganic pyrophosphatase (PPase) serves as a catalyst for the hydrolysis of pyrophosphate into inorganic phosphate, which is required for cellular phosphate metabolism and energy metabolism [18]. Under certain conditions, such as attenuated respiration, inorganic pyrophosphate (PPi), a stable and easily storable high-energy compound, can substitute for ATP in glycolysis-related reactions [19]. The involvement of PPA1 in the generation of mitochondrial membrane potential is induced by PPi [20]. Therefore, the significant negative regulation of PPA1 in the LF group can likely be attributed to insufficient energy supply, resulting in decreased sperm motility.

In normal physiological processes, oxidoreductase activity is indispensable, as it maintains normal sperm function. This function is accomplished through the production of reactive oxygen species (ROS). However, an imbalance between the production of ROS and the protective effect of the antioxidant system can lead to oxidative stress [21]. Despite this potential harm, ROS is still essential for various sperm functions, including sperm maturation, sperm overactivation, sperm capacitation, and the acrosome reaction [22]. We hypothesize that the freezing process may cause oxidative stress damage to sperm due to the excessive production of ROS, resulting in a decrease in sperm vitality and functional impairment [23,24]. This aligns with previous research studies that indicate that oxidative stress may be associated with the enrichment of the PDIA4 and PDIA6 pathways [25]. PDIA proteins, known as redox folding catalysts, play a crucial role in maintaining redox balance and signal transduction in the endoplasmic reticulum. Additionally, they function as molecular chaperones [26]. One member of the PDIA family, PDIA4, is prominently distributed in the sperm head, affecting sperm maturation and coordinating intracellular redox states [27]. PDIA4 is also involved in maintaining redox homeostasis in various cells, including mouse embryos and cancer cells [28,29]. Similarly, PDIA6, another oxidoreductase, interacts with heat shock protein A2 (HSPA2) to regulate cell–cell adhesion and maintain the reductive status of the plasma membrane [30]. We speculate that an increased expression of PDIA6 may protect HSPA2 from oxidative stress, thus safeguarding sperm [31]. In experimental conditions, the up-regulation of PDIA4 and PDIA6 expression in the HF group may potentially repair the oxidative stress damage caused by the freezing process, although further research is required to confirm this hypothesis. Furthermore, in our study, we observed a significant down-regulation of FTH1 (ferritin heavy chain 1) following the freeze–thawing procedure of goat spermatozoa, suggesting an association with the ferroptosis pathway. FTH1 is an iron storage protein complex that plays a crucial role in maintaining intracellular iron balance and protecting cells from oxidative stress caused by free iron overload [32]. Previous studies have shown that promoting FTH1 expression inhibits the accumulation of ferrous ions and lipid ROS in cells, indicating its importance in iron homeostasis [33]. We speculate that the cryopreservation of sperm may induce ferroptosis, leading to an increase in FTH1 expression. However, the underlying mechanism remains unclear, highlighting the need for further research in this area.

This study identified the presence of chaperone proteins among the selected sperm proteins, suggesting that the testes may have more efficient basic cellular functions. These proteins could potentially impact the antifreeze resistance of sperm. HSPB1, a member of the small heat shock protein (HSP20) family, acts as a molecular chaperone that facilitates the proper folding of other proteins when exposed to environmental stress [34]. In the HF group, HSPB1 is up-regulated, potentially mitigating the effects of stress caused by sperm freezing. However, further research is needed to understand the underlying mechanism. STIP1 is a cochaperone protein that plays a crucial role in enhancing the survival ability of germ cells under high-temperature stress conditions [35]. It shares its homology with the human heat shock cognate protein 70 (hsc70)/heat shock protein 90 (hsp90)-organizing protein (Hop). DNAJB1, belonging to the DnaJ or Hsp40 (heat shock protein 40 kD) family of proteins, acts as a molecular chaperone that stimulates the ATPase activity of Hsp70 heat-shock proteins [36]. This facilitates proper protein folding and prevents the aggregation of misfolded proteins. In rodents, DNAJB1 is localized in the acrosome and tail of the sperm, suggesting a potential association with sperm motility. However, further verification is required to establish this relationship [37]. It is hypothesized that the down-regulation of STIP1 and DNAJB1 may contribute to the impaired stress response, thereby affecting the resistance of sperm to freezing.

Capacitation is a crucial process for mammalian sperm to acquire fertilization ability within the female reproductive tract. Frozen stimulation, however, can cause irreversible damage to sperm, leading to premature capacitation. This premature capacitation can result in changes in the physiological characteristics of the sperm, ultimately affecting fertilization [38,39]. ATP1A1, an integral membrane protein responsible for establishing and maintaining the electrochemical gradients of Na^+^ and K^+^ across the plasma membrane, plays a role in sperm capacitation in bovine sperm [40]. It is worth noting that Na/K-ATPase isoforms, specifically ATP1A4 in the testes and ATP1A1 ubiquitously, are abundant in the bovine sperm plasma membrane. A previous study identified ATP1A4 as a regulator of sperm motility [41]. In the present study, we observed a down-regulation of ATP1A1 in the LF group, which suggests that cryopreservation has a significant impact on goat sperm membrane-bound proteins.

ADAMs are a family of membrane-anchored proteins that are widely distributed in different species and present in various tissues. Among the ADAM genes, ADAM7 is predominantly expressed in the epididymis [42]. Previous research has shown that mouse ADAM7 is associated with calnexin (CANX), heat shock protein 5 (HSPA5), and integral membrane protein 2B (ITM2B) in sperm membranes [43]. In our study, we observed an up-regulation of ADAM7 expression in the LF group, suggesting the possible impact of freezing–thawing on sperm membrane and an increase in the number of sperm with abnormal motility. The absence of ADAM7 has previously been demonstrated to lead to reduced fertility and changes in sperm membrane integrity in mice [44]. ABCG2, an efflux pump belonging to the ABCG subfamily of ATP-binding cassette (ABC) transporters, is localized in the plasma membrane of mammalian cells [45]. Recent studies have also reported the presence of ABCG2 in spermatogonias, the acrosomal region of mouse and rat epididymal spermatozoa, as well as human and bull ejaculated spermatozoa [46]. In our study, an up-regulation of ABCG2 expression was observed in the HF group, which may be associated with sperm plasma membrane transport.

Serine/arginine-rich (SR) family proteins are widely studied pre-mRNA splicing regulators [47]. SRSF1 is a member of this family; it continuously shuttles between the nucleus and cytoplasm and is involved in post-splice activities [48]. Previous studies have demonstrated its role in spermatogenesis, and the targeted disruption of SRSF1 in mice has been shown to result in early embryonic death [49,50]. Furthermore, knockout studies in mice have demonstrated a critical role for SRSF1 in spermatogenesis [51]. These findings emphasize the important contribution of SRSF1 in spermatogenesis. Our study showed that SRSF1 expression was significantly reduced in the HF group compared to the LF group, which may impair sperm viability and affect sperm splicing, ultimately leading to decreased sperm quality. Although these studies explain the discrepancy in protein levels in cryopreserved sperm, further studies are needed to elucidate the precise underlying mechanisms.

## 4. Materials and Methods

### 4.1. Chemicals

The chemicals used in this study were all obtained from Sigma-Aldrich (Beijing, China) unless stated otherwise.

### 4.2. Semen Collection

The semen samples utilized for the research were from a group of 12 healthy Guanzhong dairy goats (2–3 years old). These dairy goats were fed the same food and provided unlimited access to water. All dairy goat specimens were found to be fertile and had their ejaculates collected three times a week. The ejaculates of each goat were pooled to make a single semen sample. They were located in Fuping County, Weinan City (Shaanxi, China), in the Aonike Ranch. 

To gather the ejaculates from dairy goats, professionals used the fake vagina technique. This procedure was used ten times, resulting in the collection of 4–6 mL of sperm per goat. A drop of the sperm sample was placed onto a glass slide that had been pre-warmed to 37 °C to assess sperm motility. After that, the material was examined with a phase-contrast microscope (Olympus, Tokyo, Japan). Sperm with a vitality greater than 75% was chosen for additional cryopreservation experiments.

### 4.3. Preparation for Semen Cryopreservation

A total of 3.1 g of glucose, 4.6 g of lactose, and 1.5 g of sodium citrate were dissolved in 100 mL of distilled water to provide a solution for the base extender of goat semen. The extender was created ahead of time before semen processing. Following full chemical breakdown, the extender was filtered via a bacterial filter membrane. After that, egg yolk (with a final concentration of 15%), glycerin (with a final concentration of 4%), and antibiotics (Penicillin Sodium and Streptomycin, at a concentration of 1000 IU/mL) were introduced into the extender. The concentration of sperm from twelve goats was measured and adjusted to 1.0 × 10^8^ spermatozoa/mL by adding the extender. Eventually, the samples were cooled to 4 °C for further examination.

The 4 °C cooled semen was separated into 0.25 mL straws and frozen at 140 °C using a programmed freezer (IMV Technologies, Digitcool 5300, L’Aigle, France) according to the following freezing curve: 4 to 10 °C for 10 min, 10 to 60 °C for 5 min, and 60 to 140 °C for 10 min. These straws were then frozen in liquid nitrogen at 196 °C. After two weeks of storage, the straws were thawed individually in a water bath at 40 °C for 30 s. After that, the samples were tested for sperm quality and antioxidant activity.

### 4.4. Sperm Samples Preparation

After a two-week storage period in liquid nitrogen at a temperature of −196 °C, the straws were individually thawed in a water bath at 37 °C for 30 s. Following the thawing process, an assessment of sperm motility was conducted. From this assessment, the top 4 sperm motility specimens were selected and categorized as the high freezability (HF) group, while the last 4 sperm motility specimens were categorized as the low freezability (LF) group. Centrifugation at 800 r/min for 5 min at 4 °C was performed on ejaculates from both the HF and LF groups to remove the seminal plasma. This centrifugation step was repeated twice, and the resulting samples were stored at −80 °C for protein extraction and proteomic analysis.

### 4.5. Assessment of Sperm Quality

To assess sperm motility and motion parameters, specifically straight-line sperm velocity (VSL), curve-line sperm velocity (VCL), average path velocity (VAP), and linearity (LIN), thawed semen (10 μL) was deposited onto a pre-warmed glass slide (37 °C). The computer-assisted semen analysis system (CASA, HVIE-SSW V8.0) was utilized for this purpose [52]. For comprehensive evaluation, five fields of view were randomly selected per group, with each containing a minimum of 200 spermatozoa. To ensure accuracy, at least three replicates were performed for all groups, and the final result was recorded as the average of these replications.

To evaluate the integrity of the plasma membrane in goat sperm, we used the SYBR-14/PI (LIVE/DEAD^®^ Sperm Motility Kit; Thermo Fisher Scientific, Waltham, MA, USA). Thawed semen (100 μL) was added to a centrifuge tube, which already had 0.1 μL of SYBR-14. The mixture was incubated at 37 °C for 10 min. Then, 0.5 μL of propidium iodide (PI) was added to the sample for staining, followed by another 10 min of incubation at 37 °C. We analyzed the sperm plasma membranes using a fluorescence microscope (ECLIPSE 80i; Nikon, Shanghai Henghao Instrument, Shanghai, China). Green fluorescence indicated sperm with an intact plasma membrane, while red fluorescence indicated damaged spermatozoa. The percentage of green fluorescent spermatozoa was used to express the plasma membrane integrity [53]. Each slide contained a minimum of 200 sperm from three different fields of view. The measurement was repeated at least three times, and the results were reported as the average of three replicates.

Fluorescein isothiocyanate-peanut agglutinin staining (FITC-PNA; Nanjing Jiancheng Bioengineering Institute, Nanjing, China) was used to measure the acrosome integrity of the goat sperm. Initially, 50 μL of semen was applied to separate the microscope slides and subsequently air-dried. To fix the cells, absolute methanol was added to the slides before they were left for 10 min. Subsequently, a solution containing 30 μL of FITC-PNA (100 μg/mL) diluted in phosphate-buffered saline (PBS) was evenly distributed across each slide. The slides were then incubated in a dark environment at 37 °C for 30 min. Afterward, the slides were rinsed with PBS and air-dried. The assessment of acrosome integrity was performed using a fluorescence microscope equipped with a Nikon DXM digital camera. The proportion of intact acrosomal sperm was determined based on the observed ratio of bright green fluorescence [54]. At least 200 sperm from three distinct fields of view were evaluated on each slide. Each measurement was repeated at least three times, and the reported final result was the average of these three replicates.

### 4.6. Analysis of Antioxidant Activity

All tests were performed according to the corresponding manufacturer’s instructions for each kit, and the test kits used were as follows: total antioxidant capacity (T-AOC) assay kit (Solarbio, Beijing, China), catalase (CAT) assay kit (Solarbio, Beijing, China), superoxide dismutase (SOD) assay kit (Solarbio, Beijing, China), reactive oxygen species (ROS) assay kit (Solarbio, Beijing, China), and malondialdehyde (MDA) assay kit (Solarbio, Beijing, China). The absorbance values of the samples at 593, 240, 560, 488, and 525 were measured using a spectrophotometer (Shanghai Spectrophotometer Co., Ltd., Shanghai, China). All the assay steps and calculation methods were performed in accordance with the corresponding manufacturers’ instructions. Each experiment was performed at least three times, and the results are reported as the average of the three replicates.

### 4.7. Protein Extraction and Trypsin Digestion

The HF and LF spermatozoa specimens were pulverized using liquid nitrogen to create cellular powder before being transferred to a 5-mL centrifuge tube. Next, the cell powder was mixed with four portions of lysis buffer (8 M urea, 1% protease inhibitor cocktail). Subsequently, the mixture underwent sonication on ice for three minutes using a high-intensity ultrasonic processor (Scientz, Ningbo, China). The centrifugation step was then performed at 12,000× *g* at 4 °C for 10 min to eliminate any remaining debris. Finally, the resulting supernatant was collected, and the protein concentration was determined using the BCA kit (Tiangen, Beijing, China).

The protein sample was initially added to a final concentration of 20% (*m*/*v*) TCA to precipitate the protein. Thorough mixing was ensured by vortexing the mixture, which was then incubated at 4 °C for 2 h. Following incubation, the precipitate was collected through centrifugation at 4500× *g* for 5 min at 4 °C. To remove any remaining impurities, the precipitated protein was washed three times with pre-cooled acetone and dried for 1 min. Next, the dried protein sample was redissolved in 200 mM TEAB and dispersed using ultrasonication. For digestion purposes, the samples were incubated overnight with trypsin, maintaining a trypsin to protein ratio of 1:50. Prior to digestion, the sample was reduced by treating it with 5 mM dithiothreitol for 30 min at 56 °C, followed by alkylation with 11 mM iodoacetamide for 15 min at room temperature in darkness. Finally, the peptides were desalted using a Strata X SPE (Phenomenex, Torrance, CA, USA) column.

### 4.8. LC-MS/MS Analysis

The tryptic peptides were dissolved in solvent A and directly loaded into a home-made reversed-phase analytical column (25 cm length, 100 μm i.d.). The mobile phase consisted of solvent A (0.1% formic acid, 2% acetonitrile in water) and solvent B (0.1% formic acid, 90% acetonitrile in water). Peptides were separated using the following gradient: 0–22.5 min, 6–22% B; 22.5–26.5 min, 22–34% B; 26.5–28.5 min, 34–80% B; 28.5–30 min, 80% B (all at a constant flow rate of 700 nL/min on an EASY-nLC 1200 UPLC system, Thermo Fisher Scientific). The separated peptides were analyzed using an Orbitrap Exploris 480 (Thermo Fisher Scientific, Waltham, MA, USA) with a nano-electrospray ion source. An electrospray voltage of 2300 V was applied. Precursors and fragments were analyzed using the Orbitrap detector. The full MS scan resolution was set to 60,000 for a scan range of 350–1400 *m*/*z*. The MS/MS scan was fixed at a first mass of 120.0 *m*/*z* with a resolution of 15,000. HCD fragmentation was performed with a normalized collision energy (NCE) of 27%. The automatic gain control (AGC) target was set at 1 × 10^6^, with a maximum injection time of 22 ms. 

### 4.9. Database Search

The DIA data were processed using the DIA-NN search engine (v.1.8). The tandem mass spectra were searched against the Ovis_aries_9940_PR_20230529.fasta (23,110 entries) concatenated with the reverse decoy database. Trypsin/P was specified as the cleavage enzyme, allowing for up to 1 missing cleavages. The fixed modifications specified were excision on N-term Met and carbamidomethyl on Cys. The FDR was adjusted to <1%.

### 4.10. Bioinformatics Methods

The eggNOG online database http://eggnog5.embl.de/#/app/home (accessed on 30 June 2023) and KEGG-mapper online tool http://www.kegg.jp/kegg/mapper.html (accessed on 30 June 2023) were utilized to perform Gene Ontology (GO) and Kyoto Encyclopedia of Genes and Genomes (KEGG) analyses of the DEPs identified between HF and LF spermatozoa. Pathways with a corrected *p*-value < 0.05 were considered significantly enriched. Protein–protein interactions (PPIs) were investigated by searching all DEPs database accession or sequence against the STRING (version 11.5) database https://cn.string-db.org/ (accessed on 30 June 2023).

### 4.11. Statistical Analysis

The data were analyzed and visualized using GraphPad Prism 8.0 software, and the results are presented as the means ± standard error of the mean (SEM). The significance level was set at *p* < 0.05, and the outcomes are expressed as mean ± SEM. A total of three biological replicates and three technical replicates were utilized for all experiments.

## 5. Conclusions

This study involved conducting a comprehensive analysis of the DEPs between HF and LF frozen–thawed dairy goat spermatozoa using a 4D-DIA-based quantitative proteomic approach. The results showed that HF spermatozoa were abundant in antioxidant enzymes and proteins associated with energy metabolism, such as NDUFB8, SDHC, and COX6C. Our functional analysis of DEPs provided insights into the molecular mechanisms underlying sperm damage caused by cryopreservation. Overall, this study serves as a valuable reference for optimizing the cryopreservation process and modifying cryogenic components.

## Figures and Tables

**Figure 1 ijms-24-15550-f001:**
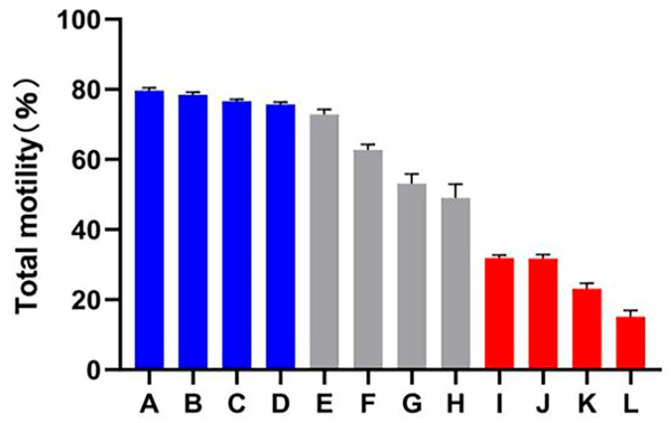
Total motility percentage of frozen–thawed spermatozoa among all dairy goats. The HF group is represented by the blue columns (A–D), the LF group is represented by the red columns (I–L), while the gray columns indicate goats whose sperm was categorized as having medium freezability (E–H).

**Figure 2 ijms-24-15550-f002:**
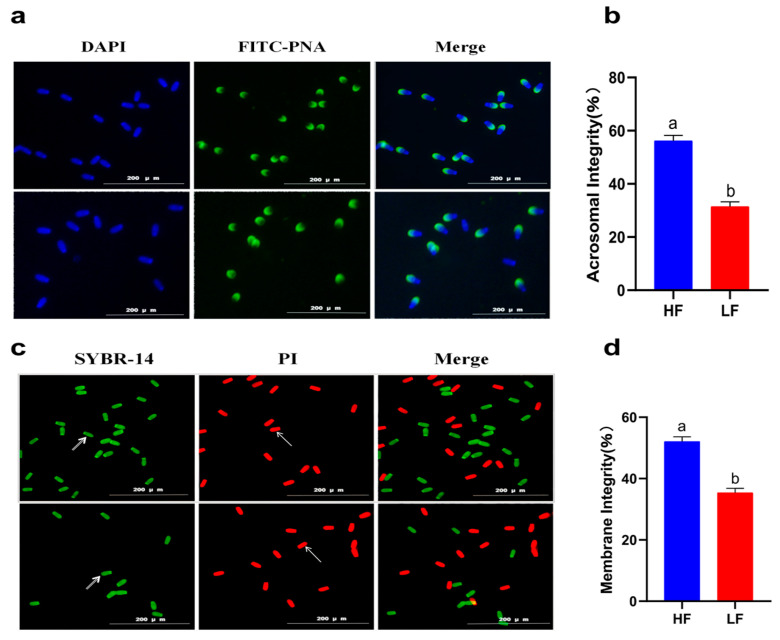
Sperm quality between the HF and LF group. (**a**) Observation of acrosome integrity. (**b**) The effect of freezability on sperm acrosome integrity. Different letters indicate significant differences (*p* < 0.05). (**c**) Observation of plasma membrane integrity. The arrows indicate that the plasma membrane is intact or damaged. (**d**) The effect of freezability on sperm plasma membrane integrity. Different letters indicate significant differences (*p* < 0.05).

**Figure 3 ijms-24-15550-f003:**
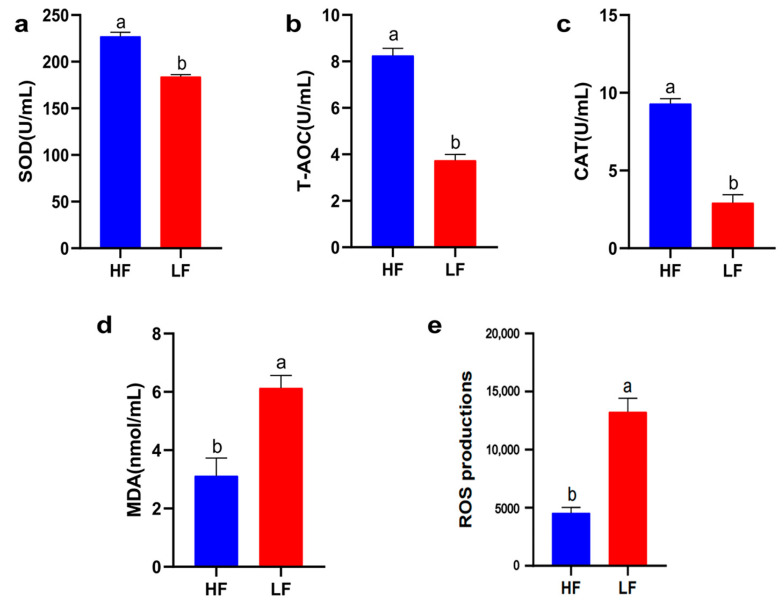
Sperm antioxidant activity between the HF and LF groups. (**a**) The effect of freezability on SOD activity in sperm. (**b**) The effect of freezability on T-AOC activity in sperm. (**c**) The effect of freezability on CAT activity in sperm. (**d**) The effect of freezability on MDA levels in sperm. (**e**) The effect of freezability on ROS levels in sperm. Different letters indicate significant differences (*p* < 0.05).

**Figure 4 ijms-24-15550-f004:**
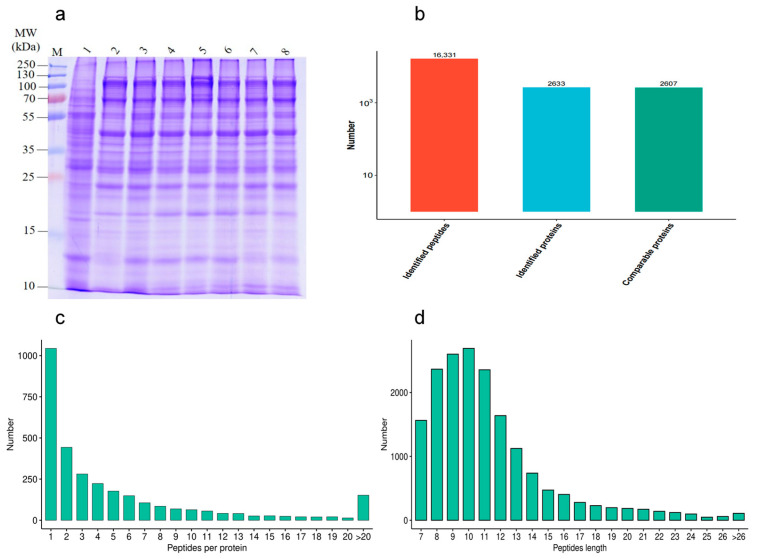
Identification of protein expression profiles between the HF and LF groups. (**a**) SDS-PAGE; (**b**) statistical diagram of mass spectrometry results. (**c**) The distribution of peptide length. (**d**) The distribution of peptide number.

**Figure 5 ijms-24-15550-f005:**
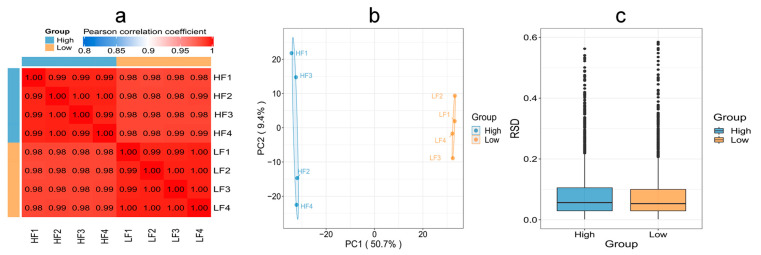
Reproducibility of protein expression between the HF and LF groups. (**a**) Pearson’s correlation coefficient. (**b**) PCA. (**c**) RSD (*p* < 0.05).

**Figure 6 ijms-24-15550-f006:**
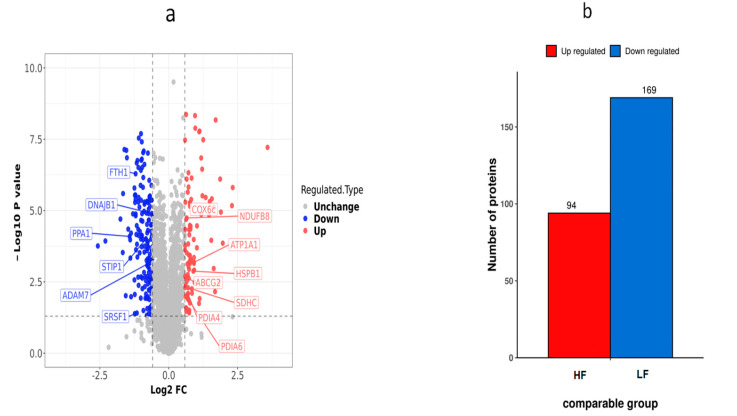
Differential protein expression distribution in the HF and LF groups. (**a**) Differential protein volcano graph in the LF and HF group; each dot in the volcanic map represents a protein. The red dots and blue dots in the figure are the proteins that are significantly differentially expressed (FC > 2, *p* < 0.05), and the grey dots are the proteins with no difference. (**b**) Histogram of the number of differential proteins in the HF and LF groups.

**Figure 7 ijms-24-15550-f007:**
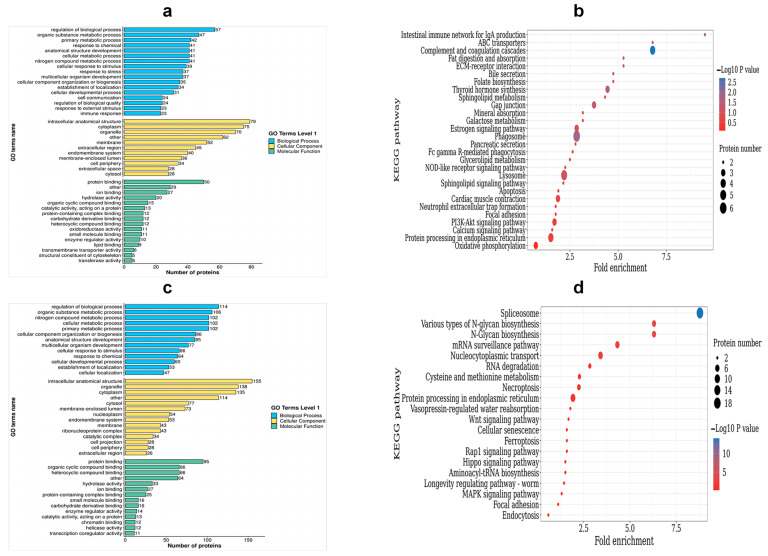
Functional enrichment analysis of DEPs. (**a**) The GO enrichment terms of the up-regulated DEPs; (**b**) the top 20 KEGG pathways of the up-regulated DEPs; (**c**) the GO enrichment terms of the down-regulated DEPs; (**d**) the top 20 KEGG pathways of the down-regulated DEPs.

**Figure 8 ijms-24-15550-f008:**
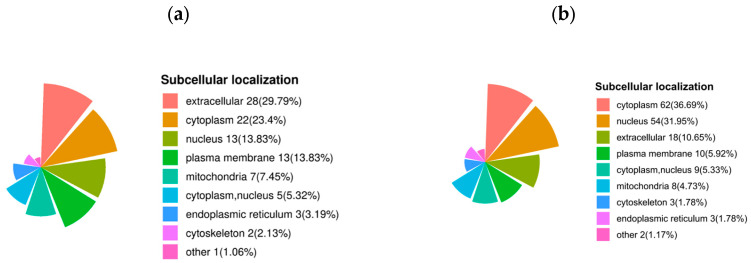
Subcellular localization of DEPs. (**a**) Subcellular localization of specific up-regulated proteins in DEPs. (**b**) Subcellular localization of specific down-regulated proteins in DEPs.

**Figure 9 ijms-24-15550-f009:**
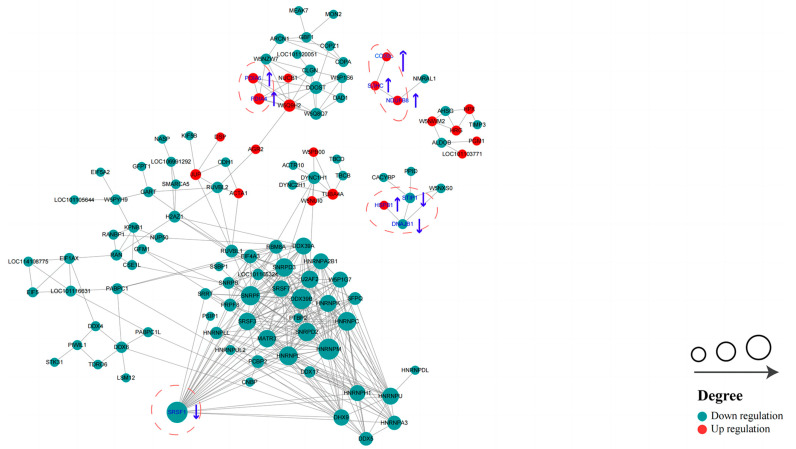
Protein–protein interaction (PPI) network of DEPs in HF and LF spermatozoa of dairy goat (created based on the STRING database). The figure presents circles that depict the DEPs, with distinct colors indicating their differential expression (dark green for down-regulated proteins and red for up-regulated proteins). The intensity of the color corresponds to the magnitude of the differential effect. Additionally, the size of each circle signifies the number of interacting proteins associated with it (The blue up arrows indicate upward regulation, the blue down arrows indicate downward regulation).

**Table 1 ijms-24-15550-t001:** Sperm motility parameters.

Groups	TM (%)	PM (%)	VAP (μm/s)	VCL (μm/s)	VSL (μm/s)	LIN (%)
HF	78.83 ± 3.21 ^a^	35.64 ± 2.62 ^a^	43.65 ± 3.12 ^a^	74.31 ± 5.32 ^a^	40.54 ± 2.64 ^a^	45.21 ± 4.21 ^a^
LF	30.21 ± 2.25 ^b^	17.45 ± 1.72 ^b^	22.31 ± 4.52 ^b^	43.12 ± 8.63 ^b^	22.36 ± 3.22 ^b^	30.31 ± 3.41 ^b^

TM, total motility; PM, progressive motility; VSL, straight-line sperm velocity; VCL, curve-line sperm velocity; VAP, average path velocity; LIN, linearity. Different letters indicate significant differences (*n* = 4, *p* < 0.05).

## Data Availability

Not applicable.

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
