# Peer review of "Proteomic Analysis of Frozen–Thawed Spermatozoa with Different Levels of Freezability in Dairy Goats"

_ijms, 2023, doi:10.3390/ijms242115550_

Round 1

Reviewer 1 Report

The authors aimed to study the effects of cryopreservation on the proteome of caprine spermatozoa.

Major issues.

The sampling conditions are not described well. First, please explain the criteria for selection of the bucks. Second, how many bucks were used and how many times were each of them sampled? This is a serious omission.

Did the authors perform a proteomics analysis BEFORE freezing the semen samples? This is not clear.

Was the proteomics analysis made in all individual samples or in pooled samples?

Please explain the Mascot analysis performed and the Gene ontology in greater detail.

Minor issues.

Please include a table to summarise the proteins found before and after freezing and please discuss these findings clearly.

Author Response

Dear Reviewer,

Thank you for your kind letter and your careful work regarding our manuscript. We have revised the manuscript in accordance with the reviewer’s comments. And point-by-point responses to the comments were as follows: 

  • Semen samples utilized in the experiments were obtained from a group of healthy Guanzhong dairy goats (2–3 years old, n = 12). These particular dairy goats received identical nourishment and had unrestricted access to water. All dairy goat specimens were confirmed to possess fertility and were subjected to semen collection three times a week and the ejaculates of each bull were combined to form a single semen sample. They were situated at the Aonike Ranch in Fuping County, Weinan City (Shaanxi, China). This change can be found in Line 352-357.
  • We didn't perform the proteomics analysis before freezing the semen samples.We performed the proteomics analysis after sperm cryopreservation. This change can be found in Section 4.3 and 4.4 .
  • The proteomics analysis was made in all individual samples.This change can be found in Section 4.8 
  • The eggNOG online database (http://eggnog5.embl.de/#/app/home) and KEGG-mapper online tool (http://www.kegg.jp/kegg/mapper.html) were utilized to perform Gene Ontology (GO) and Kyoto Encyclopedia of Genes and Genomes (KEGG) analysis of the DEPs identified between HM and LM spermatozoa. Pathways with a corrected p-value ofless < 0.05 were considered significantly enriched.Protein-protein interactions (PPIs) were investigated by searching all DEPs database accession or sequence against the STRING (version 11.5) database (https://cn.string-db.org/).This change can be found in Line 482-488.

We have revised the manuscript in accordance with the comments and marked all the amends on our revised manuscript.

Best regards.

Reviewer 2 Report

Although sperm cryopreservation is nowadays a well-managed procedure, the loss of viable gametes following thawing calls for new strategies that would help increase the proportion of viable spermatozoa suitable for assisted reproduction techniques. As such, contemporary andrology has become more focused on the search for molecular biomarkers that may predict the sperm freezability, setting up the pillars of personalized cryobiology. This article follows this line using an interesting animal model.

The experimental approach uses standard research procedures appropriate for this type of study, and the data are well-presented.

I find this manuscript interesting and original, however I have several remarks:

-          Line 37: please explain the abbreviations when used for the first time.

-          Lines 347-348: how many samples were finally used for the experiments?

-          Section 4.6.: what type of sample was used for the assays? Whole sperm suspensions or lysates?

-          Discussion: Please, discuss the limitations of the study. Add a summary or take-home message - what were the major processes the candidate proteins were associated with? Out of the candidate proteins, which could be considered as the leading ones, worth of further validation? What are prospects of the study - validation of the proteins or a different strategy?

Author Response

Dear Reviewer,

Thank you for your kind letter and your careful work regarding our manuscript. We have revised the manuscript in accordance with the reviewer’s comments. And point-by-point responses to the comments were as follows: 

Line 37: HSP90AA1, heat shock protein 90KDa alpha A1; ENO1, enolase 1; PDIA4, protein disulfide isomerase family A member 4. This change can be found in Line 38-40.

Lines 347-348: There were 8 samples used for the experiments: high freezability group (n=4) and low freezability (n=4).This change can be found in Section 4.4.

Section 4.6: The dairy goats sperm suspensions after cryopreservation was used for the assays.This change can be found in Section 4.3.

Discussion: A total of 14 candidate proteins (NDUFB8, SDHC, COX6C, PDIA6, PDIA4, PPA1, FTH1, HSPB1, DNAJB1, ATP1A1, STIP1, SRSF1, ABCG1, ADAM7) may involve several major processes, such as oxidative phosphorylation, ferroptosis, oxidoreductase activity and sperm capacitation etc.The NDUFB8, SDHC, COX6C, FTH1 and HSPB1 could be considered as the leading ones, which were worth of further validation.The prospect of the study was to provided insights of changes in the proteome and the mechanisms involved in sperm freezability for further studies in sperm cryopreservation. This change can be found in Section 3, Line 208-212.

We have revised the manuscript in accordance with the comments and marked all the amends on our revised manuscript.

Best regards.

Round 2

Reviewer 1 Report

The authors have improved the manuscript and have addressed all the concerns raised.

The manuscript needs extensive and significant improvement with regard to linguistic issues.

There are a lot of problems, which are far beyond the corrections to be made by the publishers during proof-setting stage. 
The manuscript requires complete and detailed revision from a proficient language editor.

Author Response

Dear Reviewer,

Thank you for your letter and your careful work on our manuscript. We have revised the manuscript in accordance with the comments and marked all the amends on our revised manuscript.

We hope that our revised manuscript meets your requirements. If any further action is needed, please let us know immediately.

We look forward to hearing back from you.

Best regards